



# Sea-ice thickness on the northern Canadian polar shelf: A second look after 40 years

Humfrey Melling [1]

[1]Fisheries and Oceans Canada, Institute of Ocean Sciences, PO Box 6000, Sidney Canada V8L 4B2

*Correspondence to*: Humfrey Melling (Humfrey.Melling@dfo-mpo.gc.ca)

**Abstract.** This paper presents a systematic record of multi-year sea-ice thickness on the northern Canadian polar shelf, acquired during the winter of 2009-10. The data were acquired by submerged sonar positioned within Penny Strait where they measured floes drifting south from the notional "last ice area". Ice was moving over the site until 10 December and fast thereafter. Old ice comprised about half of the 1669-km long survey. The average old-ice thickness within 25-km segments

of the survey track was 3-4 m; maximum keels were 12-16 m deep. Floes with high average draft were of two types, one with interspersed low draft intervals and one without. The presence or absence of thin patches apparently distinguished aggregate floes comprised of sub-units of various ages and deformation states from units of more homogeneous age and deformation state. The former were larger and of somewhat lower mean thickness (1-5 km; 3.5-4.5 m) than the latter (400-600 m; 6.5-14 m). Calculated ice accretion onto the multi-year ice measured in autumn 2009 was used to seasonally adjust

the observations to a date in late winter, when prior data are available. The adjusted mean thickness for all 25-km segments with 4 tenths or more old ice was 3.6 m (sample deviation of 0.4 m), a value indistinguishable within sampling error from values measured in the same area during the 1970s. The recently measured ice-draft distributions were also very similar to those from the 1970s.

**Crown copyright**

## 1 Introduction

Bourke and Garrett (1987) prepared the first compilation of sea-ice-draft (viz. thickness) surveys in the Arctic Ocean. Their map showed clearly the progressive increase in the average draft of sea ice from 1-2 m on the Siberian side of the Arctic to




more than 6 m at some places near the northern coastlines of Greenland and the Canadian Archipelago. This spatial pattern of ice thickness in the Arctic has been attributed to the combined influences of the long freezing and short thawing seasons at
high latitude (Maykut and Untersteiner, 1971) and of pack-ice convergence onto these windward shorelines (Tucker et al., 1979); the convergence maintains high ice concentration and enhances the formation of thick ice rubble via material failure under pressure (Kovacs, 1972).

A compact area of multi-year sea ice has persisted on the North American side of the Arctic during the last quarter century while the area covered by multi-year ice in the Arctic Ocean has decreased considerably (e.g. Comiso, 2012; Kwok, 2018).
Moreover the numerical models informing the 5th IPCC Assessment report suggest that the same geographic distribution will continue into the future (Collins et al., 2013; Huard and Tremblay, 2013). This evidence has fostered the concept of a future remnant "last ice area" within and adjacent to the north Greenland and Canadian polar shelves (Pfirman and Tremblay, 2009; Pfirman et al., 2010; Hamilton et al., 2014); this area is thought to be ecologically important as a future refugium for ice-reliant Arctic wildlife from bacteria to megafauna (http://www.wwf.ca/conservation/arctic/lia/ [Nov 2021]).
Although clearly of concern, the notion of a threatened last area of ice, now rapidly thinning, is based on remarkably few data from that area, past or present (Lyon, 1984; Melling, 2002; Lindsay and Schweiger, 2015). In the compilation by Lindsay and Schweiger (2015) of Arctic ice thickness observations acquired since 1975 there are points since 2003 that lie within the north-west part of the area of most persistent ice, but unfortunately this region lacks comparative data from earlier times. Moreover the recent data from this area are best viewed cautiously because they are measures of ice-plus-snow
elevation above an infrequently glimpsed sea level, from an Earth satellite with a relatively large footprint (20-70 m diameter); in situ validation has been minimal (Kwok et al., 2020).

Information on sea ice is also sparse within the adjacent part of the notional last ice area that overlies the Sverdrup Basin – the geological name for the northern Canadian polar shelf. The extent of ice here and its composition by ice type have been monitored and charted by the Canadian Ice Service since about 1970, to provide guidance for navigation during August and
September. The drift of ice through the area has been determined sporadically by following identifiable floes in satellite images and by satellite tracking of beacons on floes that fortuitously entered the area. Knowledge of ice thickness is based solely on bore-hole measurements, weekly single-point values near Canadian weather stations since the 1950s and on survey transects of bore-hole measurements acquired during on-ice seismic surveys during the 1970s. There are in addition a handful of targeted surveys of extreme features. Available data have been reviewed by Melling (2002). More recent
publications have been focused on ice kinematics and concentration, including fast ice (Galley et al, 2012; Howell and Brady, 2019), systematic estimates of ice inflow from the Arctic Ocean via feature tracking in satellite images (Alt et al 2006; Kwok 2006; Agnew et al 2008; Wohlleben et al., 2013), decadal fluctuations in multi-year ice concentration (Howell et al 2008, 2010, 2013, 2015; Tivy et al 2011) and summertime deterioration of multi-year floes in the area (Dumas et al 2007; Howell et al 2009).



Figure 1 depicts the northern part of Canadian polar shelf, annotated with the names of its larger islands and waterways. The basin is an area of consistently high ice concentration, with old ice comprising 85-90% of the ice present, except in the area north of

Bathurst Island (70%) and in Norwegian Bay (50%) (Melling, 2002). Some of this ice enters from the Arctic Ocean in the north-west during the few months of mobility in late summer and autumn; this ice tends to be highly deformed and extremely thick. A second

population develops within the basin from areas of first-year ice that do not melt during the brief summer (Flato and Brown 1996). All ice drifts generally southward to the North West Passage, entering this

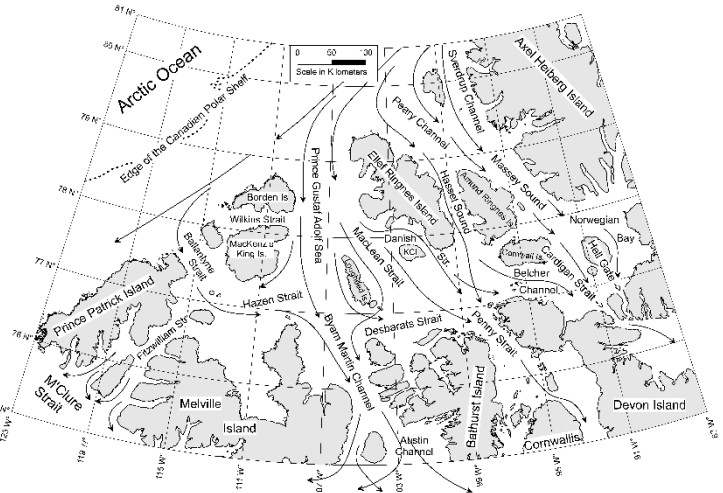

**Figure 1. The northern part of the Canadian polar shelf, its principal geographic features and the schematic pattern of sea-ice movement across it.**

waterway chiefly via Byam Martin Channel, Penny Strait and Cardigan Strait in the lower right quadrant of the map. These

channels draw ice from the Arctic Ocean principally at four entry points, Sverdrup Channel, Peary Channel, Prince Gustaf Adolph Sea and Ballantyne Strait, with a combined width of 314 km. Ice exits channels variously via Hazen, Maclean or Desbarats Straits, or Hassel or Massey Sounds, depending on wind and on ice congestion. Although ice drifts quite slowly along these pathways, it moves faster through the narrower exits to Parry Channel (combined width of 76 km), averaging about 15 km/d within Byam Martin Channel and 25-30 km/d within Penny and Cardigan Straits. Nonetheless a typical

journey across the basin takes 3-5 years because the ice is land-fast between the High Arctic islands for up to ten months per year. The time taken to cross the basin varies with conditions of ice congestion, wind and temperature in summer.

The average thickness of sea ice tends to decrease towards the south-east. In the 1970s, survey-line means as high as 5-7 m were measured near the north-western margin of the polar shelf and as high as 3-4 m near the exits to the North West Passage (Fig. 2). Melling (2002) proposed that the cause of this progressive thinning with distance from the outer shelf was a

small sustained heat flow to the ice from the ocean; this mechanism that has been further documented by Melling et al. (2015).

We do know that there were appreciable reductions in the multi-year ice coverage of the Northern polar shelf in 1961, 1970-71, 1979-81, 1983-85, 1997-99, 2005-07, 2011-12, 2014-15 and 2019 (Dumas et al., 2007; Howell et al 2010; https://iceweb1.cis.ec.gc.ca/IceGraph/page1.xhtml?lang=en [Nov 2021]). Within a few years of each occurrence, the

coverage has returned to typical historical values via recruitment of second-year ice from remaining first-year ice and via inward drift of old ice from the Arctic Ocean (Howell et al. 2015).

Future severe ice conditions on the Canadian polar shelf are critically dependent on whether all multi-year ice disappears from adjacent areas of the Arctic Ocean; cut off the supply and the waters between the Queen Elizabeth Islands could be a





very different place. It is useful to note that palaeo-
climatic indicators reveal that there was heavy ice in
this area in summer even during the Holocene Warm
Period, 5000 years ago (Dyke et al 1996; Dyke and
England 2003), when Arctic climate was 2-3°C
warmer than that during the 20th century.

The thickness of multi-year ice presently over the
northern polar shelf is not known; there have been no
observations since the late 1970s. Seeing the declining
ice thickness of the central Arctic (Lindsay and
Schweiger, 2015) one might anticipate that multi-year

ice over the northern polar shelf has also thinned
dramatically. If so, the feasibility of exploiting the
proven natural gas reserves of the Sverdrup Basin
would be greatly enhanced. Severe ice conditions here

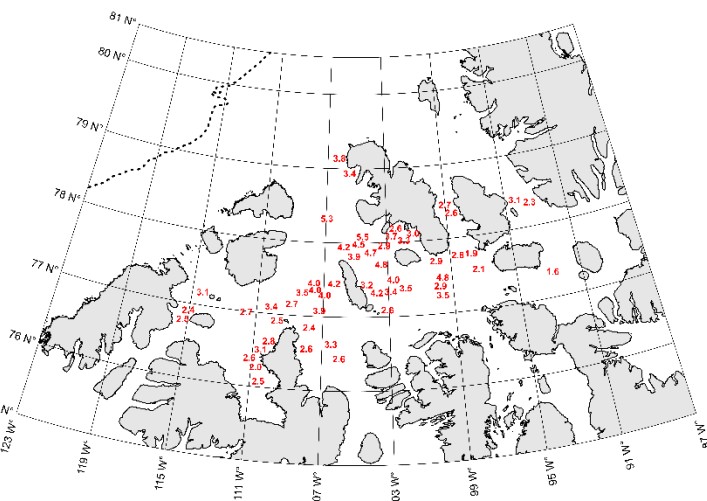

**Figure 2. Mean thickness of sea ice in late winter, 1971-1980, derived from holes drilled during seismic surveys (from Melling 2002). Data from a median of 1835 boreholes from a 2D grid of survey lines contribute to each sample. Boreholes were spaced systematically along survey lines, at least 110 yards apart (maximum of 880 yards).**

in the 1970s were critical in limiting expeditious supply of the exploration activities, and of great concern in any plan to
develop the reserves and to ship out produced and liquefied natural gas by tanker.

There is also motivation from the perspective of ice dynamics. A range of factors appear influential in the loss of ice from
the central Arctic – the pattern of ice circulation, reduced residence time, albedo feed-back, warming ocean, warming
atmosphere, poorer recruitment of second-year ice. The seas of the Canadian High Arctic are relatively isolated from the
Canada Basin, but they do harbour some of the heaviest multi-year ice in the Arctic. Study of ice here, where gain and loss
factors for sea ice may be different, could provide new perspectives on sea ice and climate change.

The goal of this study was a timely second assessment of ice within the notional last ice area. Perhaps paradoxically, we
chose not to work in the area itself but rather to observe ice as it was leaving via channels along its southern margin. The
reason, as common in Arctic field science, was logistical. Access to the interior of the Sverdrup Basin would have been very
costly by icebreaker and perhaps not practical even in summer. Use of aircraft in winter would have been cheaper but
contingent on the presence of first-year ice suitable for landing ski-planes, which is scarce here. These factors provided a
strong motivation for making observations at the most southerly useful locations. We chose Penny Strait for the first year of
observation with an eye further west on Byam Martin Channel in future years. Because inter-annual variation was likely to
be appreciable, we planned to sustain observations over several years.



## 2 Year-round ice monitoring in Penny Strait, 2009-10

Drilling or electromagnetic induction radar (Haas et al., 2009) can provide ice thickness transects at good (better than 25 m) spatial resolution but require human presence; these methods are impractical for year-round sea-ice observation at isolated locations. We chose to use ice-profiling sonar (IPS), a submerged up-looking sonar optimally designed for measuring under-ice topographic profiles. The IPS can operate autonomously for as long as three years (Melling et al., 1995). Moored at a fixed location, it measures the submerged depth of ice (draft) as the difference between the depth of the instrument (from a

pressure measurement) and its distance from the ice bottom (from an echo). The IPS is used in combination with Doppler sonar (ADCP) which measures ice-drift velocity; together the instruments provide data from which a detailed topographic transect of multi-year ice floes, ridge keels and leads can be constructed. The spatial separation of survey points is proportional to ice-drift speed, so that when ice becomes immobile the IPS is limited to measuring the slow local growth and decay of ice stalled above it.

These instruments were placed on a submerged mooring (Fig. 3) with the ADCP near the seabed and the IPS at 45-m depth, safe from moving ice. Because there was no suitable geomagnetic reference for ice-drift direction at this location – the geomagnetic inclination was 88.3° - the section of the mooring beneath the ADCP was torsionally rigid (i.e. no twisting). All ice-drift measurements by the ADCP were made relative to its fixed, but unknown, heading that was later deduced during the processing of data from the directions of measured tidal currents. This

method has been used successfully since 1998 for oceanographic moorings in Cardigan Strait, Hell Gate and Nares Strait (Münchow and Melling, 2008).

The mooring was placed in Penny Strait, on the western side that is favoured by multi-year-ice leaving the northern polar shelf;  the south-moving stream

of 4-9 tenths' pack ice on this side that was our observational target can be seen in the late-summer ice-concentration map from the Canadian Ice Service (Fig. 4).

Ironically these same streams pose challenges for the deployment and recovery of moorings. We were able to deploy our mooring from an

icebreaker (CCGS Henry Larsen) within this fast moving heavy ice. However for recovery, the ship needs to reach and hold station at the pre-determined location of the mooring, requirements that would have been difficult under the conditions encountered at deployment. We retrieved the mooring in late May 2010, operating from the surface of fast ice accessed by

aircraft. Details of the data collection are summarized in Table 1, Table 2 and Table 3

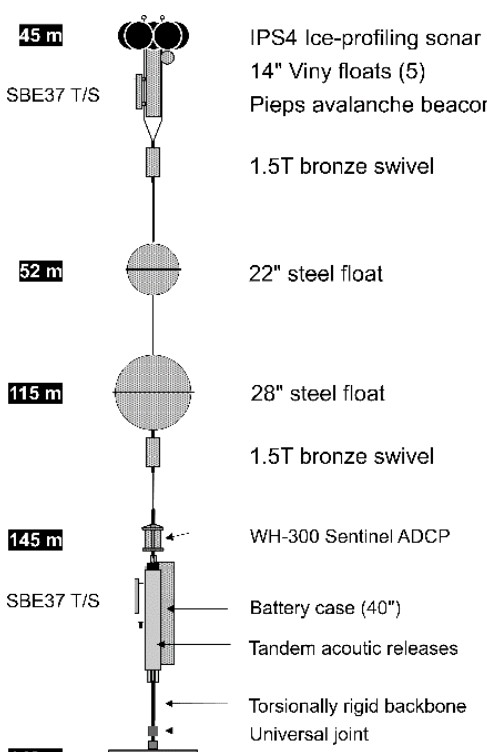

**Figure 3. Mooring used to support sonar in Penny Strait. The rigid bottom enables measurement of ice-drift direction.**



# 3 Observations

## 3.1 Environmental conditions during the period of observation

Figure 5 is a radar satellite view of Penny Strait just before the ship's arrival in 2009; it clearly shows a dense stream of
multi-year floes drifting down from the north at this time. When the ship reached the site of the installation, a team from
Canada's National Research Council completed a borehole survey on one of the floes nearby; as reported by Johnston
(2019), the minimum bore-hole thickness from 20 drillings of this floe, 1500-m across, was 8.4 m and the median was
16.1m.

The ice continued to stream southward for over a
month until October 3 when a giant floe (multi-year
ice embedded in second-year ice) from Maclean Strait
contacted land on both sides of Penny Strait and
blocked it. Ice to the north of this floe locked up on
October 25. Meanwhile multi-year floes south of the
blockage drifted out of Penny Strait, leaving an area
of young sea ice that thickened to thin first-year ice
over time. Between November 16 and 23, the
blocking floe shattered, allowing large fragments of
multi-year ice to drift into the strait. However the ice
arch behind the previously land-locked floe held,
facilitating a progressive re-establishment of fast ice
that had spread over the mooring by December 21 and
prevailed there until winter's end.

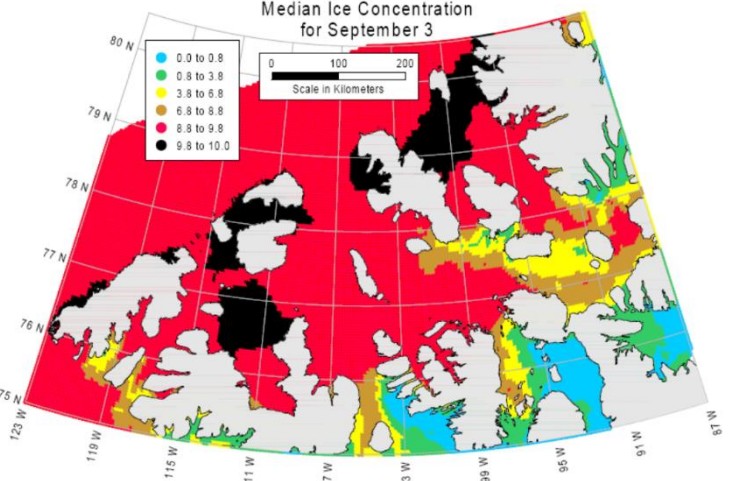

**Figure 4. Median sea-ice concentration at minimum extent in early September (data from the Canadian Ice Service). The fast-moving ice streams leaving the high Arctic appear in brown and yellow, denoting less compact ice fields more readily entered by ice-breaking ships.**

Ice conditions in Penny Strait early in 2010 are shown in Fig. 6. The ice bridge of early winter is marked by the curved line
of transition between a bright area with distinct floes (old ice) and a dark area which is level first-year ice. Old ice in smaller
floes is present to the south-east of Penny Strait, but the mooring site itself is fortuitously beneath an area of level first-year
ice. Such ice simplified access to the site by aircraft and access to the mooring submerged beneath it.





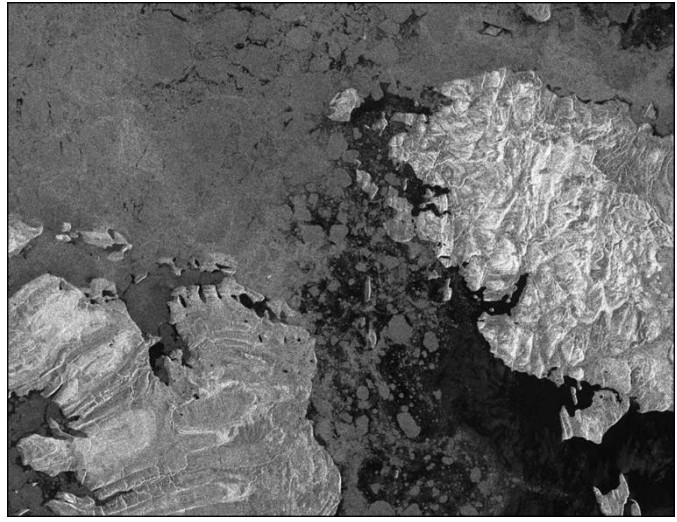

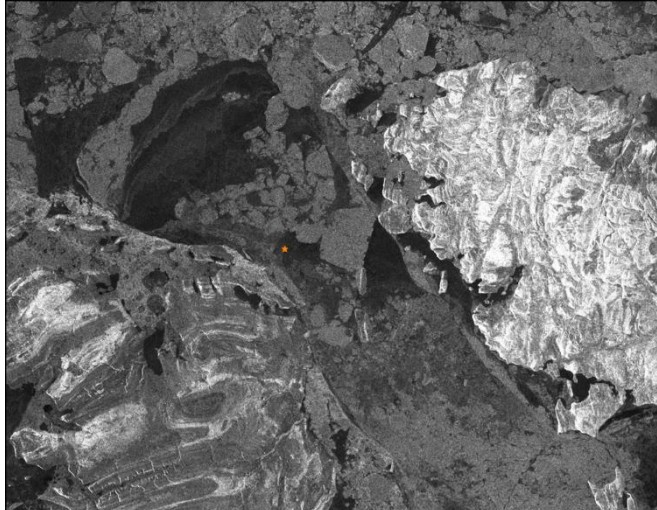

**Figure 5. Multi-year ice streaming through Penny Strait on 27 August 2009 viewed by space-based radar (RADARSAT Data and Products © MacDonald, Dettwiler and Associates Ltd. (2010) – All Rights Reserved. RADARSAT is an official mark of the Canadian Space Agency).**

**Figure 6. Ice in Penny Strait, winter 2009-10. First-year ice appears dark. The star marks the sonar (RADARSAT Data and Products © MacDonald, Dettwiler and Associates Ltd. (2010) – All Rights Reserved. RADARSAT is an official mark of the Canadian Space Agency).**

Ice and ocean responses to wind in a channel depend upon wind speed and its direction relative to the channel and on the strength of pack ice. Ice strength depends in turn on ice salinity and temperature and on the fractions of open water and thin
ice within the pack. The limiting circumstance, which occurs every winter in this area, is 10 tenths ice held immobile by coastal geometry. With fast ice established after mid-December, the wind could move neither ice nor water. In September and again in November, the direction of prevailing wind was north-west, along strait, moderate in speed in September but weak in November. During October and December, wind blew from the north-east, across the strait, moderate in October and strong in December.

**3.2 Ice drift with wind and current**

Sea-ice drift was derived via Doppler analysis of sound scattered by ice back to the ADCP when operating in bottom-tracking mode (Melling et al., 1995). Ice tracking was independent of current measurement by the same instrument, except for the determination of direction. The fixed heading of the sonar throughout the deployment (ensured by the torsional rigidity of its mooring) was determined by aligning the ebb and flow directions of the measured tidal current with those
predicted by a tidal model (https://www.bio.gc.ca/science/research-recherche/ocean/webtide/index-en.php [Nov 2021]). Sampling error, which dominates uncertainty in ice velocity measured by the Doppler method, was better than 0.5 cm s$^{-1}$ for a 15-minute average value with our operating configuration.

The progressive vector of ice drift over the mooring, approximating the ice-drift trajectory through the strait, is displayed in Fig. 7. The net displacement over 102.5 days of movement in 2009 was 529 km towards the south-east, roughly the same
distance as that between Penny Strait and the edge of the continental shelf in Prince Gustaf Adolf Sea (450 km). However



because the northern entrance to Penny Strait was blocked after December 14, much of the ice subsequently drifting over the mooring was recently formed new and young types.

The 1669-km total length of the survey track was appreciably greater than the net displacement because it continues to increase during reversals in direction driven by tide and wind. Sub-tidal events in ice movement

were an alternating sequence of excursions to the north-west and south-east. Drift to the north-west was generally brief and correlated with south-east wind (as measured at Resolute Bay): August 30 to September 3;

September13-16; September 30 to October 3;

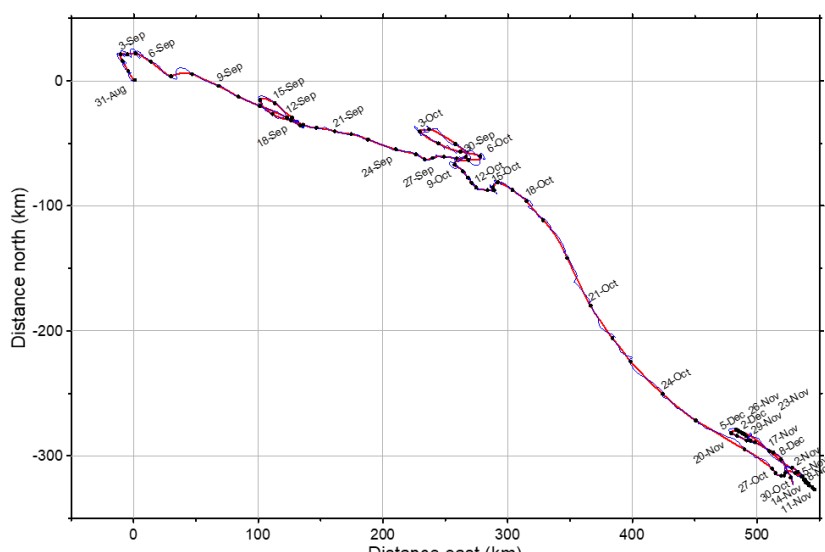

**Figure 7. Progressive vector of ice drift in Penny Strait from late August until mid-December 2009. Dots on the de-tided (red) curve are at daily intervals. Ice above the mooring became fast on December 9.**

October 6-8. Drift to the south-east occurred during the intervening periods, except that ice was almost motionless during November 5-15. Most of the net south-east displacement occurred with prevailing north-west wind during September 5-28 (251 km) and October 15-27 (374 km).

## 3.3 Draft and underside topography of sea ice

The draft of sea ice was derived from IPS' measurements of water pressure and echo time (Melling et al., 1995). Each draft value was assigned a position along the progressive vector calculated by integration of ice velocity. Because the IPS's field of view is about 0.8 m, redundant data were eliminated by re-sampling the ice-draft at 1-m increments along the drift vector. The resulting quasi-spatial transect of under-ice topography, 1.669 million data points over the 1669 kilometres, is the starting point for scientific interpretation in this study.

Statistics of ice draft for each 25-km segment of the drift path are displayed in Fig. 8. Although sonar measures draft not thickness, axes showing thickness have been added. A 1.13-ratio of thickness to draft has been used for the minimum and mean values, based on the reasonable assumption of near-isostatic balance at scales of 100 m and more (Melling et al, 1993). This ratio is consistent with densities of 1025 kg m-3 for seawater, 910 kg m-3 for submerged ice and 870 kg m-3 for sub-aerial ice (Timco and Frederking, 1996).

Because ridges and keels are not locally in isostatic balance, the ratio of thickness to draft is variable on small (3-30 m) scales in deformed ice (Hibler 1972; Melling et al., 1993). Nonetheless the ratio of maximum draft to maximum height is commonly used as a measure of the thickness of ice ridges, even though the ridge crest may not exactly coincide with the keel.   In first-year ice, the ratio is quite high; Kovacs (1972) and Sisodiya and Vaudrey (1981) report 4.5 and 5.5



respectively. The ratio is lower for multi-
year ice where thermal weathering and
consequent consolidation affect keel draft
more than sail height (Amundrud et al
2006). Reported ratios of maximum draft
to maximum height for multi-year ice
features are 3.1-3.3 (Cox, 1972; Kovacs
et al, 1973; Kovacs et al, 1975), although
some old ridges have yielded values as
large as 5.6 (Dickens and Wetzel, 1981).
The equivalent ratio of thickness to draft
is about 1.3. The 12-16 m maximum
drafts typical of multi-year ice features
seen in 2009 (Fig. 8) could be indicative
of ice as thick as 16-21 m. However, a
conservative 1.2-ratio has been used for
the left-hand axis on the top frame of that

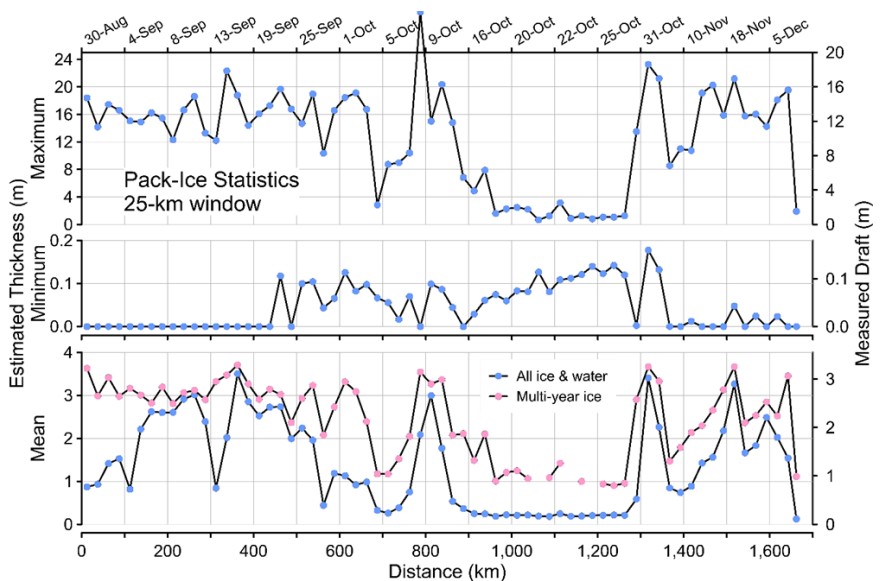

**Figure 8. Statistical measures of ice draft for each 25-km segment of the drift path. The curve labelled "multi-year ice" represents the mean for ice with draft over 75 cm. Axes at the right indicate draft and those at the left estimated thickness; conversion factors of 1.13 have been used for the mean and minimum drafts and 1.2 for the maximum, as discussed in the text. Dates of measurement are indicated on the top frame.**

figure. A keel deeper than 25 m was measured on October 8.

The curve labelled "multi-year ice" in the bottom frame represents the mean for ice with draft exceeding 0.75 m; the overall value of this metric was 2.66 m (3.0-m thickness). Those 25-km segments for which the overall mean thickness and the thick-ice thickness are close in value are presumed to have been dominated by multi-year ice. The minimum, mean and maximum values of draft enable a tentative classification of pack ice over the site as follows: Old ice in open water at varying concentration until km:450 and in thin ice until km:675; old ice in thin ice again from km:800 to km:875; new ice without old floes from km:875 to km:1275; old ice in thin ice from km:1275 to km:1350 and again from km:125 to km:1650.

The distinction between the multi-year and young-ice populations is clear in the distribution of pack-ice volume versus draft (Fig. 9). These are shown for the entire survey track and for the 25-km segments with draft greater than 2.5 m and less than 1.5 m. The distribution for all ice has a peak near zero thickness, a peak near 3-m thickness and a long

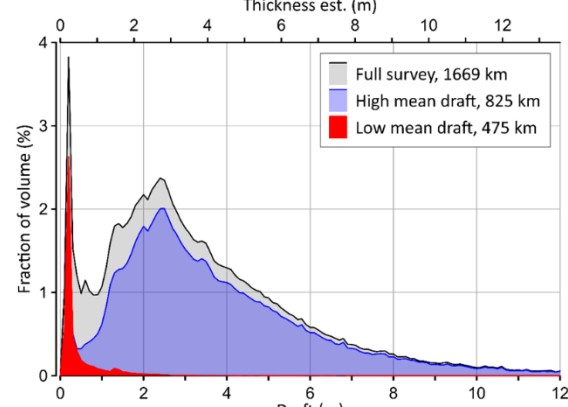

**Figure 9. Fraction of ice volume versus draft. The curves represent all data, data from 33 25-km segments with mean draft over 2.5 m and data from 19 with mean draft under 1.5 m. Thickness on the top axis is estimated as 1.13x draft..**



tail to beyond 20 m. The curve for segments with high mean draft shows clearly that the peak at 3-m thickness and the long tail are associated with old ice at this time of year, while the curve for low mean draft is clearly linked to young ice that is only lightly deformed.

## 4 Discussion

### 4.1 Evidence for change in multi-year ice of the Canadian High Arctic

As mentioned earlier, the only prior data on sea-ice thickness in this area were acquired during on-ice seismic surveys during the 1970s and discussed by Melling (2002). The surveys were conducted in the late winter (mid-March to mid-May) when the ice was thick enough to support the sled trains used to move personnel, accommodations and equipment around this remote area. There is value in comparing the 1970's data to the present observations. However because the present data were acquired in autumn and the 1970s data in late winter, a comparison is informative only after adjustments that reflect the seasonal cycle of ice growth and melt. The adjustment of ice thickness from late summer to late

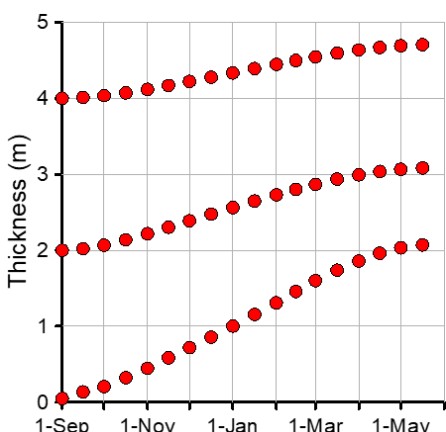

**Figure 10. Calculated wintertime growth of sea ice of three initial thicknesses. The air temperature and snow depth that control the growth are averages of climatological conditions at Resolute Bay and Eureka.**

winter is more tractable than the converse, because the rate of ice thickening by freezing is better understood than its thinning by melting. Moreover, it is reasonable to assume negligible ice thickening via ridging in the area north of Penny Strait because islands limit the build-up of wind stress and inhibit the ice movement needed to build ridges during most of the winter.

Ice growth by freezing is driven by the temperature difference between the snow or ice surface and the ocean and limited by the rate at which heat can be conducted through the ice and snow; the rate of growth slows as the ice and snow thicken. We have calculated wintertime ice growth under High Arctic conditions using 15-day averages of climatological data for air temperature and snow depth at Resolute Bay and Eureka to drive a steady state ice-growth model; assumed values of thermal conductivity were 2.03 W/m/K for sea ice and 0.31 W/m/K for snow. Results are shown in Fig. 10 for ice that was 0, 2 and 4 m thick on September 1. First-year ice starting at 0 m has thickened by 2.1 m in mid-May, whereas 2-m ice has added only 1.1 m, 4-m ice only 0.7 m and 8-m ice only 0.35 m.

Ice-draft histograms measured in autumn can be adjusted to late winter simply by shifting the bin boundaries to reflect growth by amounts calculated according to the growth model. An estimate of the late winter average thickness can then be computed by summing the histogram-weighted contributions from each of the adjusted thickness bins.

Figure 11 shows estimates of the average thickness reached by the ice in each 25-km segment by the end of the 2009-10 winter. Averages are shown for the entire ice field and for those parts of the ice field deemed to be multi-year ice by virtue of





thickness – more than 0.75 m in the autumn and more than 2.3 m in the

spring. This figure can be compared with Fig. 8 showing the observations of the ice field in the preceding autumn. The average thickness that the observed ice field might have attained by the spring of

2010 is 2.89 m (draft 2.56 m), with a standard deviation of 0.7 m among 25-km samples. The thickness of the fraction deemed old ice in the fall (draft more

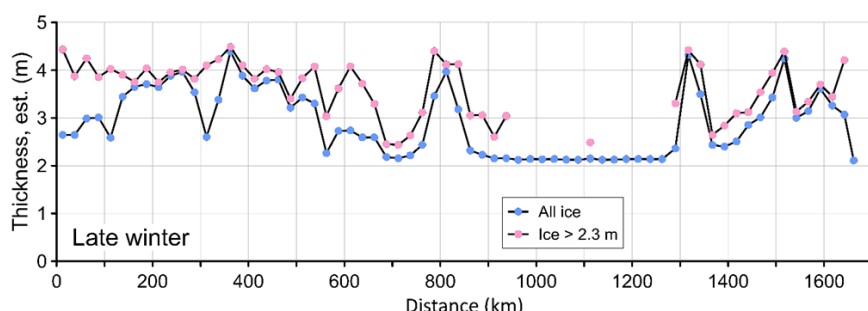

**Figure 11. Average ice thickness for 25-km segments of the track line at the end of the 2009-10 winter, estimated by forcing ice growth with monthly climatological data for air temperature and snow depth, with the assumption that no new leads or ridges formed during the winter.**

than 0.75 m) could have reached a thickness of 3.82 m (draft 3.38 m) by spring.

This number is of limited value for comparison with historical data because the pack ice diverges (Fig. 3) as its drift accelerates towards Penny Strait. This is the reason why the concentration of multi-year ice measured in Penny Strait in 2009 was lower than in the areas to the north-west, which is where the surveys of the 1970s were completed. In 2009, multi-year ice dominated only 40% of the 25-km segments measured whereas the median old-ice concentration in the areas of comparison was 70-80% (https://iceweb1.cis.ec.gc.ca/30Atlas/page1.xhtml?grp=Guest&lang=en [Nov 2021]).   A mean

thickness based on those 25-km segments that were at least half populated by multi-year ice is a better metric for comparison. This mean is 3.6 m with a sample standard deviation of 0.4 m.

Figure 12 compares track-mean values of thickness from systematic drilling in the area north-west of Penny Strait during winters in the 1970s (Melling 2002) with the range of values estimated for the same time of year in 2010, based on the 2009 measurements by sonar. The perhaps surprising result is that the estimated average thickness of multi-year sea ice up-drift of

Penny Strait in 2010 was essentially the same as it was four decades earlier in the 1970s.

Figure 13 compares the distribution of ice volume with ice thickness from the 1970s' bore-hole data and from the 2009 measurements by sonar, adjusted to late winter values. The distributions derived from bore-hole are 'noisier' than the one

derived from sonar because the number of data is so much smaller. Nonetheless it is clear that modal values were near 3-m thickness both in the 1970s and in 2009, and the roll-offs of the distributions with increasing thickness are much the same presently as in the past. Again the evidence suggests minimal

change in the multi-year ice population here during the last four

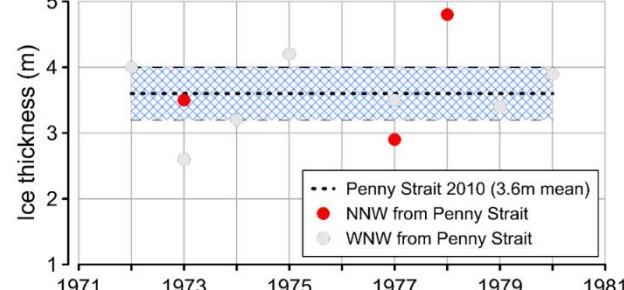

**Figure 12. Track-mean ice thickness of multi-year ice from drill-hole surveys north-west of Penny Strait during late winter in the 1970s, compared with 2010 values derived from observations in the autumn of 2009, as explained in the text. (shaded band is ±σ).**



decades.

Whereas a single year's data cannot provide the final word on change on the Canadian polar shelf, the apparent lack of change in 2009-10 relative to the 1970s is remarkable. During this time the

perennial ice cover of the central Arctic Ocean had decreased by half (Perovich et al. 2014) and the average thickness of the mix of ice types there had decreased by an estimated 1.7 m (Kwok and Untersteiner 2011).

Discussion of Arctic perennial ice in recent years has been focused

more on loss factors (ice export from the Arctic Ocean, duration of residence there, duration of the thaw season, albedo feedback, cloud effects, etc.) than on formative aspects. Yet the sea ice presence within the Arctic actually represents the difference between competing processes of ice formation and loss. The

classic perspective on multi-year ice formation is thermodynamic

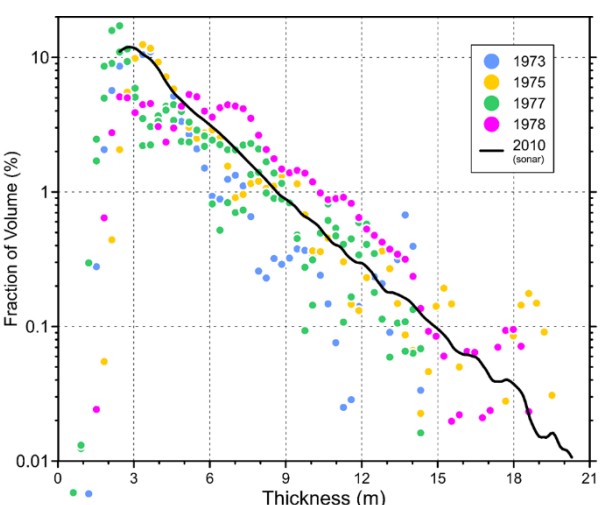

**Figure 13. Distribution of pack-ice volume over ice thickness in Penny Strait comparing recent data with surveys during the 1970s. The 2009 data have been adjusted to late winter 2010, as explained in the text.**

and traceable to the one dimensional heat-budget model of Maykut and Untersteiner (1971). Using this tool, it can be shown that first-year ice in the colder parts of the marine Arctic can survive its first melt season and gradually evolve into multi-year ice if the cold-season accretion of ice via latent heat loss through the top surface exceeds the warm-season ablation caused by absorption of radiative and oceanic heat. Because the latent heat lost from the top of the ice must diffuse through it

from the bottom, its flux is inversely proportional to ice thickness whereas heat gains during the thaw season are proportional to surface area not thickness. As the ice grows thicker, the amount of new-ice accretion during winter decreases; a quasi-equilibrium maximum thickness is reached when winter's gains match summer's losses. This maximum is about 3 m for an average seasonal cycle of climatic forcing in the High Arctic (Maykut and Untersteiner 1971; Flato and Brown 1996). Calculations show that increasing the maximum above 3 m by varying climatic forcing within observed limits is difficult,

but that increases in snow depth and ocean heat flux can readily decrease it.

Paradoxically a large fraction of the perennial ice mapped for the 1970s by Bourke and Garrett (1987) was appreciably thicker than the 3-m thermodynamic limit. Even then it was obvious that this much thicker ice must have formed by a different process. Narrow zones of much thicker ice (ridges) formed via the piling of fracturing of level ice are ubiquitous in pack ice (Wadhams and Horne 1980). Indeed ridged ice ten or more times thicker than the fragments that comprise it forms

within a few hours during wind storms (Amundrud et al. 2004). Because subsequent thermal deterioration of such thick ice amounts to a few metres per year, remnants of a 30-m ridge may persist for a decade or more (Amundrud et al. 2006). However because ridge remnants are separated and relatively narrow (15-50 m) they cannot form the extensive areas of multi-year ice above 3-m thickness mapped by submarine.



However, the stamukhi zone forming the interface between fast and mobile ice is a plausible source for the large floes of
thick multi-year ice, past and present. Here cyclic offshore/onshore and shearing movements of the pack driven by storms
facilitate the creation of young ice in flaw leads and its subsequent compression into broad expanses of very thick ice rubble
near the grounding line (cf. Kovacs and Mellor 1974). The patterns of wind and ice circulation generate the highest ice
pressure and rubble-forming potential along the North American side of the Arctic margin (Thorndike and Colony 1982;
Colony and Thorndike 1984), particularly against northern Greenland and the western Canadian Arctic Archipelago. This
inexorable compression against the coast is evident (Fig. 14) in the drift-pattern of ice revealed by satellite-tracking (IABP,
2020), which displays large along-shore movements in both directions that amount to relatively little net displacement along-
shore, interspersed with smaller perpendicular movements that are shoreward on average.

During the 12 to 18-month duration of each track, ice drifted shoreward at about 450 m per day – actual values were, from
north-east to south-west, 360, 435 and 555 m per day. Since floes are very closely packed in this area, the shoreward
movement is possible only if intervening ice is removed from the sea surface, likely via ridge-building in response to
compressive and possibly shearing forces at the fast-ice edge (or coastline). If the "lost" ice were 2 m thick, each day's
movement would be sufficient to build along the coast
a ridge of 20-m draft, 30-degree side slope and 25%
void fraction. In the course of a year, such "daily"
ridges, packed together, would form a stamukhi zone
almost 30 km wide and an average of 11 m in
thickness. With time, the cold climate of this region,
its highly compacted ice cover and sluggish drift
towards milder climes almost guarantee the
transformation of broad bands of stamukhi into thick
floes of multi-year ice.

The formation and consolidation of stamukhi into
broad multi-year hummock fields is storm driven,
rapid and insensitive to the direct effects of
atmospheric warming. Because the rate of ablation
under present conditions is far too slow to eliminate

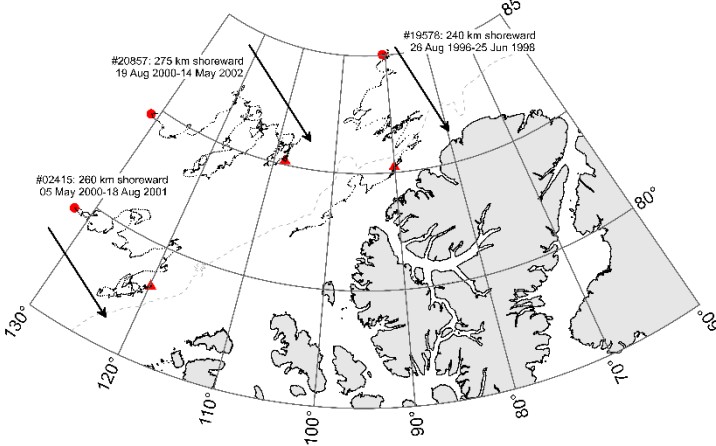

**Figure 14. Tracks of 3 beacons on ice north-west of the Canadian Arctic Archipelago, illustrating the combined shearing and compressive deformations that allow the ice slowly to approach the coast. The starting and ending points are marked by red dots and triangles, respectively. Data courtesy of the International Arctic Buoy Program (IABP)** http://iabp.apl.washington.edu/index.html accessed Nov 2021

the resulting thick floes in one thaw season, it is plausible that such ice persists in areas close to the formations zones with
properties similar to those of past decades. We suggest that thick-ice genesis as stamukhi explains, despite changing climate,
the similarity between our 2009 ice thickness data and those from the 1970s in the Canadian High Arctic.




## 4.2 Hazardous ice

High average draft (correlated with high thickness and strength), wide extent and the absence of thin (viz. weak) zones are all attributes of interest in the context of ice hazards to navigation. This data set has

been analysed to yield geometric data (draft, extent) on large hazardous features within multi-year ice of the High Arctic in the late 2000s. The focus was on floes wider than 500 m (big, vast and giant in standard nomenclature: CIS, 2005) which might be detectable using the 250-m resolution of the satellite-borne sensors commonly used for ice

reconnaissance. These multi-year floes defy simple characterization (e.g. Johnston and Timco, 2008) because they are commonly freeze-bonded conglomerations of smaller floes of varying provenance, age and thickness; moreover, thaw holes produced beneath melt ponds in summer may confound even a reliable identification of floe boundaries

in topographic sections. To facilitate both the delineation of floes and comparisons with bore-hole surveys, we have worked with a 250-m

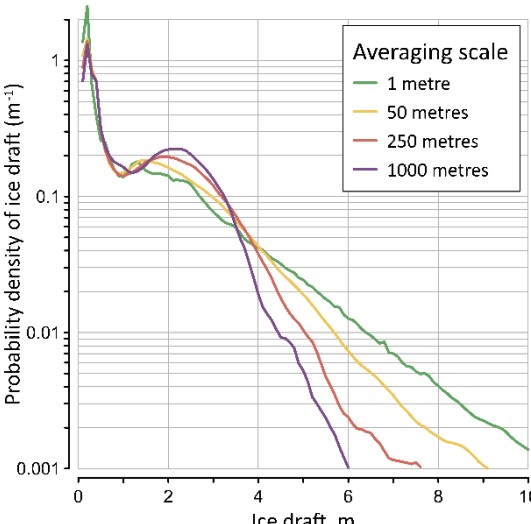

**Figure 15. Impact of along-track averaging on the probability density of ice draft.**

running average of the sonar-derived draft profile. Figure 15 illustrates how averaging alters the probability density of ice draft: the likelihood of both thin and thick ice decreases as the averaging scale increases while the likelihood of near-average ice increases; the multi-year-ice mode becomes more prominent and shifts to higher draft because the ice-draft histogram is

positively skewed. These differences are important to bear in mind when interpreting data from low resolution sensors carried by aircraft and satellites.

The Rayleigh criterion was applied to this smoothed ice-bottom relief to detect expanses of ice that were unusually thick on a 250-m scale; the reference level for draft, -2 m, enabled identification of the thickest parts of thick floes; peaks in draft were counted as separate features if intervening values dropped towards this reference level by more than half the peak value. This

is an unorthodox use of the Rayleigh criterion, otherwise long established as a tool for identifying ridge keels in well resolved ice-bottom topography (e.g., Wadhams and Horne, 1980). The boundaries of floes containing these features were taken to be those locations where the 250-m averaged draft crossed 1.5-m depth. The length of the measured chord on each floe was taken to be the length of the survey path between these cross-overs; the mean draft of each floe was the average of all draft values in the measured (not smoothed) data between these points.





Figure 16 displays a segment of the smoothed ice-bottom relief, the "keels" identified within that segment and the edges of the floes in which they were embedded. These so-called "keels" are actually broad expanses of thicker ice, and despite their similarity in a vertically exaggerated view, they not associated with individual pressure ridges (Fig. 17).

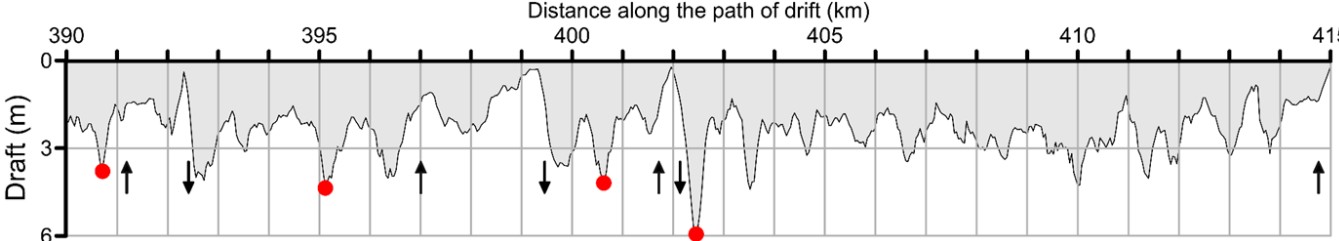

**Figure 16. A 25-km segment of the section measured by sonar, The curve displays 250-m averaged values of ice draft. The red dots mark "keels" identified using the Rayleigh criterion. The down and up arrows mark the arriving and departing edges of floes, respectively.**

Figure 18 displays the distribution along the survey path of floes potentially hazardous to navigation. The hazard is represented by an index of severity equal to the area of the floe's cross-section associated with ice thicker than 4 m.

Although thick ice and large floes both present hazard, the index does not distinguish between short transects of very thick ice and long transects of thinner ice. Table 4 compares the 10 floes with the highest severity and the 10 thickest floes. The two groups of floes differ by an order of magnitude in chord length (high index floes being larger) and by 50% in mean thickness (high index floes being thinner), but do not differ significantly in 1-m and 250-m scale maximum thickness. The scatter-plots in Fig. 19 showing all 177 floes suggest that the maximum 250-m averaged thickness of each floe had a weak

tendency to increase with the size of the floe and a strong tendency to increase with its mean thickness. Within the full sample of 177 floes (plot not shown), the mean thickness of floes did not vary systematically with floe size.

The data herein provide an opportunity to estimate how frequently floes such as those carefully measured by Johnston (2011, 2019) were likely to be encountered in the Canadian High Arctic in the late 2000s. Autonomous sonar is impartial in measuring all floes that pass overhead, whereas bore-hole surveys are selective. Johnston sought suitable floes during

reconnaissance flights by helicopter and selected the most rugged for systematic measurement via drilling. She worked in the

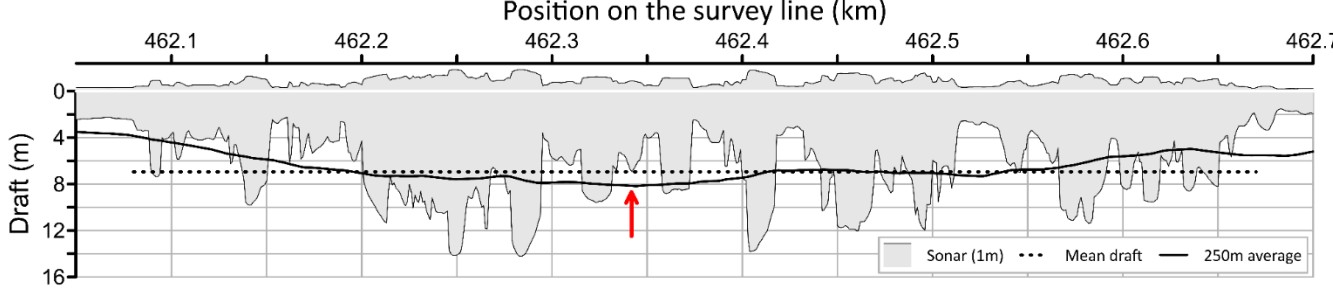

**Figure 17. A 650-m section of a floe with 6.95m average draft showing the difference between conventional ridge keels of 25-m scale in the original sonar record and the identified "keel" (arrow) in the 250-m averaged draft. Plotted freeboard is simply a notional scaled version of the floe's draft.**





same area of the High Arctic during an overlapping period of time (August 2007 to April 2011). The mean of the floe-averaged thicknesses from 22 floes, each typically 20-30 holes at 10-m spacing (a 200-300 m span), was 7.60 m; the quartiles were 4.95, 7.60 and 9.38 m.

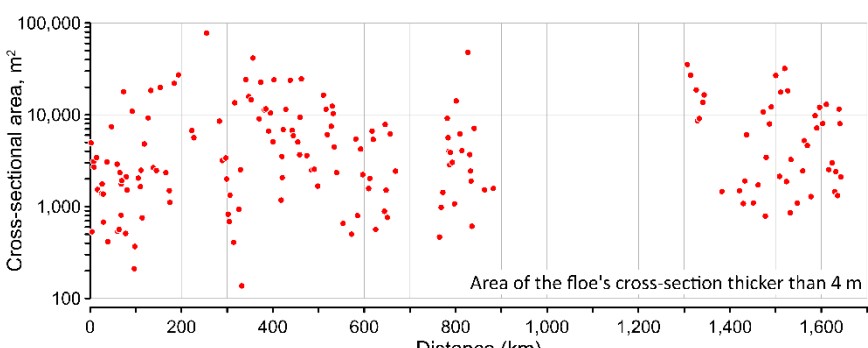

**Figure 18. An index of the navigational hazard posed by individual thick floes identified along the survey path. The index in the cross-sectional area of all ice thicker than 4 m in each floe.**

The sonar in Penny Strait measured ice draft across 1669.0-km of pack ice, a mix of thick old ice, thin new ice and open water. There were 738.7 km of ice within the section that had draft greater than 0.95 m. Within this thicker component of the pack, a 250-m expanse of ice with draft exceeding 4.35 m was encountered on average every 5.7 km, one exceeding 6.70 m every 42 km and one exceeding 8.30 m every 113 km (these thresholds are the drafts equivalent to quartile thicknesses of the bore-hole surveys). As to be expected given the drillers' specific interest in very thick ice, 250-m expanses of ice with draft exceeding 4.35 m were much less commo n in the sonar record (4.36%) than in the drill-hole surveys (75%), as also those exceeding 8.30 m (0.22%, versus 25%).

However, the recurrence distances for thick ice estimated from the sonar record are not inconsistent with the airborne observer's success in finding such features. An observer flying along the sonar section at 100 knots and looking only down would cross a 250-m expanse of 9.4-m thick ice (8.3 m draft) ice on average every 36 minutes; an observer with visibility a few kilometres to each side of the flight path could find such features in less time. Search times in the range of 15-30 minutes are consistent with the experience of the drill team (M. Johnston, pers. com., 2009).

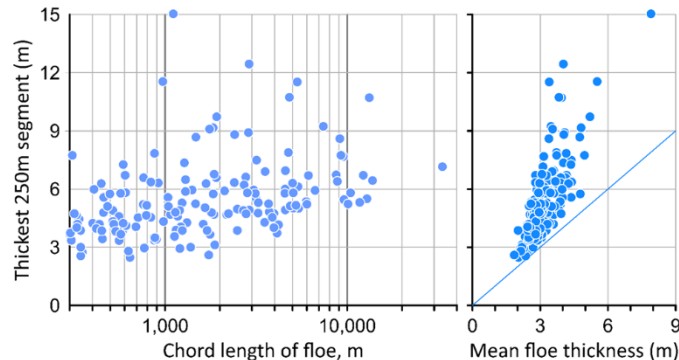

**Figure 19. Scatterplot of the maximum 250-m averaged thickness of each floe against the measured chord length of that floe (left), and against the mean thickness of the floe (right).**

## 5 Conclusions

Ice-profiling sonar installed in Penny Strait during the 2009-10 winter have provided modern ice-thickness data from the northern Canadian polar shelf, for comparison with prior bore-hole surveys during the 1970s. The two data sets provide a perspective on change within the notional last ice area of the marine Arctic.

The ice pack in Penny Strait was a mix of multi-year floes from further north and young ice formed locally. The latter occupied about 60% of the pack ice surveyed because the densely packed south-drifting multi-year floes spread apart as they



accelerated into the strait. Old ice could be distinguished from new ice the two populations formed distinct modes in the histogram of draft. The mean draft of the component of the pack that was deemed to be old (draft > 0.75 m) was 2.66 m
(thickness 3.00 m).

The observations in 2009 were acquired at the end of the thaw season, whereas comparative data from the 1970s were acquired late in the freezing season. To allow inter-comparison, the modern ice-draft data were converted to thickness and then seasonally adjusted for wintertime ice accretion. The resulting mean estimated thickness for multi-year in May 2010 was 3.6 m (σ=0.4 m). The average thickness from ten bore-hole lines measured between 1972 and 1980 within 100 km to the
northwest of Penny Strait was also 3.6 m (σ=0.6 m). Moreover, the 2010 distribution of ice volume as function of thickness was not obviously different from that in the 1970s in terms of their modes and roll-offs with increasing thickness.

The apparent lack of change in thick multi-year ice near Penny Strait after 40 years of Arctic warming suggests examining its genesis not its wastage. We propose that the formation of Arctic multi-year ice floes thicker than about 3 m has been and continues to be driven by pack-ice pressure rather than by surface thermal energy balance. The build-up of pressure
sufficient to build large ridges occurs within a few hundred kilometres of coastlines during strong onshore wind storms. The occurrence of such weather is not obviously reduced by rising Arctic air temperature.

Where such winds prevail, specifically off northern Greenland and the Canadian Archipelago, ridges accumulate into wide bands of thick ice rubble – stamukhi zones. Ice rubble piled to a thickness of tens of metres takes many years to deteriorate in the cold High Arctic climate. Meanwhile it evolves into the very thick old ice still found along this coast and for a few
hundred kilometres in down-drift directions across the Canadian polar shelf and into the Beaufort and Greenland Seas.

The northern Canadian polar shelf continues to present serious ice hazards to navigation. 177 floes of interest were identified within the 739 km of ice having draft greater than 0.95 m. Their danger to navigation was assessed using 4 metrics: chord length, maximum 250-m-average thickness, maximum thickness and the cross-sectional area of ice thicker than 4 m. The 20 most dangerous floes fell into two groups based on these metrics, those with the most ice thicker than 4 m and those with the
highest mean thickness. The first tended to have modest mean thickness (3.4-m group average) and large chord length (12.5-km group average); the second had much smaller chord length (1.3-km group average) and, obviously, high mean thickness (5.1-m group-average). The groups differed little in the greatest thicknesses on 1-m and 250-m scales. 46 of the floes has at least one 250-m expanse with average thickness over 6 m of floes and 14 had an expanse thicker than 8 m. The thickest 250-m expanses tended to occur within the smaller thicker floes.

**Data availability statement**

Information for access of data: https://www.pac.dfo-mpo.gc.ca/science/oceans/data-donnees/index-eng.html [Nov 2021].

The archive of Fisheries and Oceans Canada is not accessible via the internet. Data used in this study are available on request to the corresponding author. Users may also contact the senior analyst to arrange for access:

DFO.PAC.SCI.IOSData-DonneesISO.SCI.PAC.MPO@dfo-mpo.gc.ca



**Author contribution**

The manuscript is the work of the sole author.

**Competing interests**

The author declare that he have no conflict of interest..

**Acknowledgements**

This study was funded by the Canadian Program for the International Polar Year, the Canadian Program for Energy Research and Development and the Government of Nunavut. Logistical support was provided by the Polar Continental  Shelf Program (PCSP) of Natural Resources Canada and the Canadian Coast Guard. I thank PCSP staff, officers and crew of CCGS *Henry Larsen*, and DFO technical staff (R. Lindsay, J. Poole) for their support in mooring preparation, deployment and recovery. I thank D. Riedel and staff of ASL Environmental Sciences Inc. for diligence in processing sonar data.

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


**Tables**

**Table 1. Oceanographic mooring in Penny Strait**

| Site name | ACW09-8 | |
|---|---|---|
| Position (WGS84) | 76° 40.5019' N | 097° 52.5356'W |
| Water depth | 148 m | |
| Deployment platform | CCGS Henry Larsen | |
| Deployment date | 30 Aug 2009 | |
| Recovery platform | Sea ice, via aircraft from Resolute Bay | |
| Release date | 18 May 2010 | |


**Table 2. Ice-measuring instruments on the Penny Strait mooring**

| | Ice Profiling Sonar | Acoustic Doppler Current Profiler |
|---|---|---|
| Model | ASL Environmental Sciences Ltd. IPS4 | Teledyne RDI Workhorse-300 |
| Measurements | Ice draft from echo range & pressure | Ice drift & current velocity from Doppler |
| Depth | 44 m | 145 m |
| Sampling interval | 1, 3, 10 s for Sep-Oct, Nov-Jan, Feb-May | 15 minutes |
| End of in-water record | 01:20 utc   18 May 2010 | 09:45 utc   16 Mar 2010* |

**Table 3. Characteristics of hazardous multi-year ice features.**

| | 10 floes with the most ice thicker than 4 m | | | | | | 10 floes with the greatest mean thickness | | | | |
|---|---|---|---|---|---|---|---|---|---|---|---|
| Severity index ($m^2$) | Chord length (m) | Mean thickness (m) | Thickest 250-m (m) | Maximum thickness (m) | | Mean thickness (m) | Chord length (m) | Severity index ($m^2$) | Thickest 250-m (m) | Maximum thickness (m) |
| 78,019 | 33,756 | 3.1 | 7.2 | 17.9 | | 7.9 | 1,110 | 9,156 | 15.0 | 29.7 |
| 47,879 | 13,226 | 3.9 | 10.7 | 19.6 | | 5.5 | 995 | 5,627 | 11.5 | 23.6 |
| 41,529 | 13,893 | 3.4 | 6.4 | 18.0 | | 5.2 | 1,921 | 11,548 | 9.7 | 17.2 |
| 35,390 | 9,377 | 3.7 | 7.7 | 16.7 | | 5.0 | 421 | 2,486 | 7.7 | 14.5 |
| 31,916 | 8,828 | 3.7 | 6.4 | 15.1 | | 4.8 | 1,837 | 9,395 | 9.2 | 18.9 |
| 27,218 | 10,678 | 3.2 | 5.8 | 14.8 | | 4.8 | 536 | 3,423 | 5.8 | 10.0 |
| 26,981 | 9,109 | 3.4 | 8.6 | 17.2 | | 4.8 | 1,512 | 8,011 | 8.7 | 18.8 |
| 26,761 | 10,135 | 3.1 | 5.2 | 15.1 | | 4.4 | 842 | 4,075 | 6.4 | 14.4 |
| 24,630 | 7,463 | 3.5 | 9.2 | 17.1 | | 4.4 | 3,176 | 14,633 | 7.5 | 15.5 |
| 24,159 | 8,879 | 3.2 | 6.7 | 21.4 | | 4.4 | 498 | 2,701 | 5.6 | 10.5 |
| 36,448 | 12,534 | 3.4 | 7.4 | 17.3 | **Sample mean** | 5.1 | 1,285 | 7,105 | 8.7 | 17.3 |
| ±5,517 | ±2,573 | ±0.1 | ±0.6 | ±0.7 | | ±0.4 | ±285 | ±1,371 | ±1.0 | ±2.0 |






**List of Figures**

Figure 1. The northern part of the Canadian polar shelf, its principal geographic features and the schematic pattern of sea-ice movement across it.

Figure 2. Mean thickness of sea ice in late winter, 1971-1980, derived from holes drilled during seismic surveys (from Melling 2002). Data from a median of 1835 boreholes from a 2D grid of survey lines contribute to each sample. Boreholes were spaced systematically along survey lines, at least 110 yards apart (maximum of 880 yards).

Figure 3. Mooring used to support sonar in Penny Strait. The rigid bottom enables measurement of ice-drift direction.

Figure 4. Median sea-ice concentration at minimum extent in early September (data from the Canadian Ice Service). The fast-moving ice streams leaving the high Arctic appear in brown and yellow, denoting less compact ice fields more readily entered by ice-breaking ships.

Figure 5. Multi-year ice streaming through Penny Strait on 27 August 2009 viewed by space-based synthetic aperture radar (RADARSAT Data and Products © MacDonald, Dettwiler and Associates Ltd. (2010) – All Rights Reserved. RADARSAT is an official mark of the Canadian Space Agency).

Figure 6. Ice in Penny Strait, winter 2009-10. Multi-year ice is light-toned and first-year ice shows dark. The star marks the sonar (RADARSAT Data and Products © MacDonald, Dettwiler and Associates Ltd. (2010) – All Rights Reserved. RADARSAT is an official mark of the Canadian Space Agency).

Figure 7. Progressive vector of ice drift in Penny Strait from late August until mid-December 2009. Dots on the de-tided (red) curve are at daily intervals. Ice above the mooring became fast on December 9.

Figure 8. Statistical measures of ice draft for each 25-km segment of the drift path. The curve labelled "multi-year ice" represents the mean for ice with draft over 75 cm. Axes at the right indicate draft and those at the left estimated thickness; conversion factors of 1.13 have been used for the mean and minimum drafts and 1.2 for the maximum, as discussed in the text. Dates of measurement are indicated on the top frame.

Figure 9. Fraction of ice volume versus draft. The curves represent all data, data from 33 25-km segments with mean draft over 2.5 m and data from 19 with mean draft under 1.5 m. Thickness on the top axis is estimated as 1.13x draft.

Figure 10. Calculated wintertime growth of sea ice of three initial thicknesses. The air temperature and snow depth that control the growth are averages of climatological conditions at Resolute Bay and Eureka.

Figure 11. Average ice thickness for 25-km segments of the track line at the end of the 2009-10 winter, estimated by forcing ice growth with monthly climatological data for air temperature and snow depth, with the assumption that no new leads or ridges formed during the winter.

Figure 12. Track-mean ice thickness of multi-year ice from drill-hole surveys north-west of Penny Strait during late winter in the 1970s, compared with 2010 values derived from observations in the autumn of 2009, as explained in the text. (shaded band is ±σ).





Figure 13. Distribution of pack-ice volume over ice thickness in Penny Strait comparing recent data with surveys during the 1970s. The 2009 data have been adjusted to late winter 2010, as explained in the text.

670 Figure 14. Tracks of 3 beacons on ice north-west of the Canadian Arctic Archipelago, illustrating the combined shearing and compressive deformations that allow the ice slowly to approach the coast. The starting and ending points are marked by red dots and triangles, respectively. Data courtesy of the International Arctic Buoy Program (IABP) http://iabp.apl.washington.edu/index.html [2021]

Figure 15. Impact of along-track averaging on the probability density of ice draft.

675 Figure 16. A 25-km segment of the section measured by sonar, The curve displays 250-m averaged values of ice draft. The red dots mark "keels" identified using the Rayleigh criterion. The down and up arrows mark the arriving and departing edges of floes, respectively.

Figure 17. A 650-m section of a floe with 6.95m average draft showing the difference between conventional ridge keels of 25-m scale in the original sonar record and the identified "keel" (arrow) in the 250-m averaged draft. Plotted freeboard is 680 simply a scaled version of the floe's draft.

Figure 18. An index of the navigational hazard posed by individual thick floes identified along the survey path. The index in the cross-sectional area of all ice thicker than 4 m in each floe.

Figure 19. Scatterplot of the maximum 250-m averaged thickness of each floe against the measured chord length of that floe (left), and against the mean thickness of the floe (right).