# Peer review of "Thickness of multi-year sea ice on the northern Canadian polar shelf: A second look after 40 years"

_The Cryosphere, 2021_

## Referee Comment (RC1)

Review of "Sea-ice thickness on the northern Canadian polar-shelf: A second look after 40 years"

By: H. Melling

Submitted to The Cryosphere – tc-2021-393

This paper presents a unique set of observations of ice thickness collected in Penny Strait with a moored sonar during winter 2009-10, that are then compared to historic in situ observations manually collected during spring in the 1970s. The paper shows that MYI in this area did not thin over this 50 year gap, which is interesting because MYI in the Arctic Ocean has been shown to thin significantly and decline in areal extent over this time period. The result highlights the dynamic formation of this very thick type of multi-year ice through convergence of the Arctic ice pack against Greenland and the Canadian Archipelago, and its subsequent drift into and through the Archipelago. Overall the paper is well written and provides a unique glimpse into the ice pack of this fairly inhospitable area of the Canadian Archipelago. I have two larger comments and several minor edits/suggestions, but overall this is a very nice paper and I think it will be suitable for publication after these revisions.

Major comments:

1) One of the key results of the paper is that MYI within Penny Strait did not become thinner between the 1970s and 2009-10. I think this is a really interesting result and I commend the authors for working with the dataset to pull out this detail from the noise and variability of the IPS dataset. However, I think it is important throughout the paper to note that while MYI is the same thickness there is overall less MYI in the CAA. I think without this clarification the paper only tells part of the story and could be misconstrued. Given that the paper is already quite long, I don't expect the authors to add in more analysis of MYI area within the CAA, but Howell and Brady (2019) and other previous Howell papers have shown that MYI area is declining in the CAA. I think it is critical that the author here reference this in the intro and reiterate it in the abstract and conclusions. Basically, clarify that MYI within the CAA has not thinned because of its source from convergence against the CAA and Greenland, however the presence of MYI within the CAA is declining.

2) My second major ('ish) comment has to do with the adjustment of ice thickness with the thermodynamic growth model. In the methods it says that the model was run from September 1 to mid-May. The total thermodynamic growth over that time was then added to the observed ice thickness throughout fall, for which the results are presented in Figure 11. My issue is that the time period for modelled ice growth does not correspond to when the ice was observed by the mooring. For example the area of young ice that drifted over the mooring between 900 and 1,300 km (October 15 - ~29), had 6-8 less weeks of ice growth than ice of the same thickness would

have had from September 1. I appreciate that ice growth slows through the season, so the impact may be minimal, but I think this can be addressed by running the thermodynamic model at different time steps, or at least discussed in the methods as a limitation of this "bulk" correction. In addition, can the time series of snow depth and air temperature used to drive the thermodynamic ice growth model be presented either in a figure or as supplementary material? Along with this it would be suitable to note that terrestrial snow measured at Resolute may differ from snow on sea ice, particularly young ice that may have formed after some of the heavier snowfalls during early autumn.

Minor comments:
Line (L) 12: consider changing "units" to "floes", I appreciate these weren't large aggregate floes like the first type of ice, but these were still discrete floes.

L 35–36: It's not essential, but it might be worth either updating this to CMIP6, or acknowledging that CMIP6 projects similar conditions in the last ice area.

L 40-46: I appreciate in this section you are focused on observations from this region, but I think it is worth noting that Moore et al., 2019 showed negative trends in ice thickness in the last ice area from PIOMAS. From this you may even be able to add a short bit of text in your discussion section about how models may not recreate the formation of very thick ice within the rubble zones along the CAA and Greenland.

L 60: revise to "… depicts the northern part of *the* Canadian Polar Shelf".

L 77-79: Is there a reference for these drift speeds?

L 75-81: It seems like it would be suitable to note here that ice import from the Arctic Ocean into the QEI is increasing due to the transition towards a longer period of ice motion during summer (Howell and Brady, 2019).

L 102-106: I think in this sentence it is also worth noting the dramatic loss of MYI from the Arctic Ocean and its retreat to this area along the CAA and Greenland. Maslanik et al., (2011) or more recently Stroeve and Notz (2018) show the loss of MYI in the Arctic Ocean that then corresponds to the reduction in ice thickness.

L 106-110: This justification for the research is focused on oil and gas activities, but I think it would also be worth noting that MYI from this area is subsequently transported southwards to the Northwest Passage, where it affects shipping. In the short term it seems like shipping is more critical than oil and gas extraction.

L 127: Suggest adding an "an" in front of IPS.

L 135-136: revise to "… at 45-m depth, *where it was safe from deep keels*."

Figure 4: What is the time period for the median ice concentration in this figure?

L 228: Switch the "-3" to superscripts.

L 261: correct the value 125, perhaps its meant to be 1350 and continue from the last class.

L257-261 and Figure 8: Can the different ice classes be shaded in the figure so that it is more clear how they differ from each other? I believe these classes are marked on Figure 18, so it would be useful to add them here as well.

L 290: Please provide a reference for the assumed values of thermal conductivity of sea ice and snow.

L 428: revise "they" to "they are".

L 428-429 and Figure 18: Why was the value of 4 m chosen to represent a navigational hazard?

L428 – 436 and Figure 18: I don't think its an area but rather "a length of the floes cross-section associated with ice thicker than 4m". Table 3 shows the Chord Lengths of floes, which I think is what should be used in the text around Figure 18 and the figure itself.

L 431: revise "Table 4" to "Table 3".

L 455: correct "common".

L 473: Something is missing from this sentence, "… from new ice __?__ the two populations…".

L 478: Worth clarifying that wintertime ice accretion was estimated from an ice growth model. Suggest you revise to read, "… then seasonally adjusted for modelled estimates of wintertime ice accretion".

L 478: revise to "…thickness for multi-year *ice* in May 2010 was …".

L 497-499: revise to " 46 of the floes *had* at least one 250 m expanse *with an average thickness* over 6m, while 14 floes had an expanse thicker than 8 m".

L 499: I would encourage the authors to write a more impactful ending of the paper, perhaps a short paragraph commenting on the future of MYI in the CAA as the Arctic ice pack continues to decline. Will thick MYI continue to exist when the sea ice is confined to the CAA and last ice area in the not too distant future?

References

- Moore, G. W. K., Schweiger, A., Zhang, J., & Steele, M. (2019). Spatiotemporal variability of sea ice in the Arctic's last ice area. Geophysical Research Letters, 46, https://doi.org/10.1029/2019GL083722
- Maslanik, J., J. Stroeve, C. Fowler, W. Emery (2011), Distribution and trends in Arctic sea ice age through spring 2011, GRL, 38, L13502, doi:10.1029/2011GL047735.
- Stroeve, J.C., D. Notz (2018), Changing state of Arctic sea ice across all seasons, Environmental Research Letters, 13(103001, doi:10.1088/1748-9326/aade56.
* * *
David Babb
University of Manitoba

---

## Author Comment (AC1)

**General**

I have found it helpful to receive these two thorough and thoughtful of the original manuscript. I thank them.

My plans for the manuscript in response to these comments are outlined below.

**Reviewer 1**

**Major**

1. **I think it is important throughout the paper to note that while MYI is the same thickness there is overall less MYI in the CAA. I think without this clarification the paper only tells part of the story and could be misconstrued.**

   The reviewer makes a valid point that the paper's conclusions might be misconstrued. Clearly I have not made it sufficiently clear that the paper is focused specifically on ice thickness and specifically on the old-ice component of sea ice cover of the Canadian High Arctic. However as noted by this reviewer, there has been a decrease in the area of multi-year ice on the Canadian polar shelf, which signifies a decrease in the volume of MYI there, notwithstanding the present null result for thickness.

   I will modify the paper's title to reflect the focus on MYI thickness and heed this reviewer's suggestion to note and reference in the abstract, introduction and conclusions the now reduced concentration of MYI in this area.

2. **Adjustment of ice thickness with the thermodynamic growth model: In the methods it says that the model was run from September 1 to mid-May. My issue is that the time period for modelled ice growth does not correspond to when the ice was observed by the mooring.**
   Without sufficient thought, I started the ice-growth calculation on September 1 which is when the mean daily temperature drops below 0°C. Although Resolute Bay typically freezes on about the same date, the CIS charts showed no new ice in the vicinity of Penny Strait in 2009 until two weeks later. Fig. 10 shows a calculated 14 cm of growth for 2-m ice from 1 September to 15 Oct and 7 cm for 4 m ice. Therefore, I agree with the reviewer that starting date for the calculation was chosen carelessly. However, the slow growth of thick ice and the not-so-robust method of adjusting ice thickness to late winter do not justify the complexity applying a different correction for each date of observation in 2009. I propose re-calculating the correction from additional starting dates of 15 September & 15 October to the quantify the difference by winter's end of corrections based on different starting dates.

3. **Can the time series of snow depth and air temperature used to drive the thermodynamic ice growth model be presented either in a figure or as supplementary material?**

   It is probably not desirable to have these data displayed in the manuscript. They are readily available on the Environment Canada website (https://www.canada.ca/en/environment-climate-change/services/ice-forecasts-observations/latest-conditions/archive-overview/thickness-data.html ). It is probably sufficient to provide the URL, but I can plot them for inclusion as supplementary material if the reviewers consider this useful.

   **Along with this it would be suitable to note that terrestrial snow measured at Resolute may differ from snow on sea ice.**

   Yes of course. However, the snow-depth data that I use was measured on the sea ice close to the weather stations and at the same place that the ice thickness was measured each week. The long-term Canadian ice-observing program has been noteworthy in its foresight to include a snow-depth component.

**Minor**

Line 12: Consider changing "units" to "floes. A good suggestion. Thank you.

Lines 35-35: It might be worth either updating this to CMIP6.

Good suggestion. You can see that my manuscript has been awaiting completion for some time!

Lines 40-46: I think it is worth noting that Moore et al., 2019 showed negative trends in ice thickness in the last ice area from PIOMAS.

I take your point, but my purpose in this paragraph is to note the dismal lack of in situ data in the mischievously named "The Canadian Hole". Without such data, we don't really know whether PIOMAS or satellite altimeters are credible or not. I prefer not to trigger a squabble here about whether they are or not.

Line 60: I will provide the missing "the".

Line 77-79: Is there a reference for these drift speeds?

For Penny Strait, I will provide a cross-reference to fig. 7 and the associated text. I can note that the DFO data for the other two straits are presently unpublished.

Line 75-81: Ice import from the Arctic Ocean into the QEI is increasing … (Howell and Brady, 2019).

This is a relevant citation, which I will include. Thank you.

102-106: I think in this sentence it is also worth noting the dramatic loss of MYI from the Arctic Ocean and its retreat to this area along the CAA and Greenland. Maslanik et al., (2011) or more recently Stroeve and Notz (2018) show the loss of MYI in the Arctic Ocean that then corresponds to the reduction in ice thickness.

I will work these references into the noted paragraph. Thank you.

Lines 106-110: This justification for the research is focused on oil and gas activities, but I think it would also be worth noting that MYI from this area is subsequently transported southwards to the Northwest Passage, where it affects shipping. In the short term it seems like shipping is more critical than oil and gas extraction.

Yes, and I hope in the long term too. Inclusion of this point will be worthwhile.

Line 135-136: My omission of a phrase is acknowledged.

Figure 4: What is the time period for the median ice concentration in this figure?

I can add to the caption "Epoch 1969-1998". Since the features to be noted here are the exit streams, which remain unchanged, I consider it unnecessary to update to the most recent epoch, but can do so if this reviewer disagrees.

Lines 228, 261: Typos to be corrected.

Lines 257-261 and Figure 8: Can the different ice classes be shaded in the figure so that it is more clear how they differ from each other?

I will add the requested shading. Thank you.

Line 290: Please provide a reference for the assumed values of thermal conductivity of sea ice and snow.
Agreed.

Line 427: Typo to be corrected.

Lines 428-429 and Figure 18: Why was the value of 4 m chosen to represent a navigational hazard?

This choice is arbitrary because hazard in practical terms depends upon the temperature of the ice (roughly speaking, the season), the design of the vessel and how it is operated. 4 m was chosen to be appreciably higher than the 3 m linked to the 3-knot-continuous-speed capability of Class 10 icebreaking ships in summertime ice. The most powerful icebreaker in service is the Russian Artika, nominally rated at about Class 9.

I will note the basis of my choice of 4 m choice in the revised text.

Lines 428-436 and Figure 18: I don't think it's an area but rather "a length of the floes cross-section associated with ice thicker than 4m". Table 3 shows the Chord Lengths of floes, which I think is what should be used in the text around Figure 18 and the figure itself.

I apologize for my lack of clarity because the index is in fact an area. It is the sum of the integrals of thickness over distance for all segments of the floe thicker than 4 m. I will work to make this clear in the text and captions.

Lines 431, 435, 473: Typos to be corrected.

Line 478: Worth clarifying that wintertime ice accretion was estimated from an ice growth model. Suggest you revise to read, "… then seasonally adjusted for modelled estimates of wintertime ice accretion".

Good point. Thank you.

Lines 478, 497-499: Typos to be corrected.

Line 499: I would encourage the authors to write a more impactful ending of the paper, perhaps a short paragraph commenting on the future of MYI in the CAA as the Arctic ice pack continues to decline. Will thick MYI continue to exist when the sea ice is confined to the CAA and last ice area in the not too distant future?

This is good suggestion. I propose responding to it by moving the two paragraphs spanning lines 482-490 to the end of this section and reworking them to provide an opinion on the two issues of interest that this reviewer has raised.

---

## Author Comment (AC2)

**General**

I have found it helpful to receive these two thorough and thoughtful of the original manuscript. I thank them.

My plans for the manuscript in response to these comments are outlined below.

**Reviewer 2**

**Major**

**The comparison between the two periods (1970s and 2009/2010) seems to me a bit tricky … Recent multi-year ice thickness is represented by data from one-mooring site with short period (2009-2010). … I would suggest the author to provide more careful examinations to strengthen the conclusion of this study … Since the manuscript focuses on long-term changes of multi-year ice thickness, sea ice growth estimates by climatological forcing may be problematic … I suggest using specific forcing from 2009/2010.**

I share the discomfort of both reviewers about the seasonal "correction" of ice draft.

See the comments below in relation to lines 286-292, and also above in relation to the first reviewer's second comment.

**Minor**

Line 47: It would help readers if the location and area of Sverdrup Basin is annotated in Figure 1.

The domain of Fig. 1 is the Sverdrup Basin. I will modify the caption to state this.

Line 65: Bathurst Island and Norwegian Bay could be also highlighted, otherwise readers have to take time to find them in the map.

I acknowledge that it may take a little time to find labelled features. However, the geography of islands and channels in the Canadian High Arctic is complicated. Given the number of geographic features referenced in the text, I believe highlighting would make things less clear, not more so.

Line 93: Queen Elizabeth Islands could be also annotated in Figure 1.

The domain of Fig. 1 covers the QEI. Therefore I will modify its caption to state this. See also my response to comment #1

Line 93-94: I agree that the meaning of "cut off the supply and the waters between the Queen Elizabeth Islands could be a very different place" is obscure.

I will revise the sentence to "The ice regime of the Sverdrup Basin would be very different if in-drift of Arctic Ocean ice were to cease".

Fig. 3: As recommended, I will switch to SI for the weights and measures on this illustration.

Line 143-145: I suggest to mark the position of the mooring in Figure 1 or another large-scale map. I would suggest to show this in a closer map showing bottom topographic features, e.g., maps covering the area shown in Figure 5.

The exact position of the mooring in Penny Strait is already marked on fig. 6; albeit, the symbol used is too small. I will also add the mooring's coordinates to the text in section 2. In reality, awareness that the mooring is on the western side of Penny Strait is sufficient in the context of this paper. As for seabed topography, it is largely irrelevant to this discussion because sea-ice drift atop highly stratified Arctic waters is little influenced by it. Moreover, an additional large format figure (in an already lengthy paper) would be required to display the complicated terrain of Penny Strait.

Figs. 5 & 6: I would suggest to show coastline on top of the image, if possible.

I understand the difficulty of distinguishing land from ice if not familiar with the geography. I will experiment with a coastline overlay to see if discrimination can be improved. I will also enlarge the red star marking the mooring on these figs.

Line 252: Why 0.75 m is used as a threshold to identify multi-year ice?

As evident in fig.9, a draft of 0.75 m marks the low-point in the histogram for all ice measured, both old and seasonal. This low point is the logical choice for this discriminant. I will make this clear in the text.

Line 257-261: Is it possible to annotate these features in Figure 8, which helps reader to understand the temporal changes of ice type.

Yes. I can readily delineate the various segments using background shading on the plot.

Line 286-292: Why has the author applied climatological data to derive the ice growth rate? I suppose that a calculation using the data from 2009-2010 could provide more accurate estimate that takes into account specific condition during the observation period.

I share the discomfort of both reviewers about the seasonal "correction" of ice draft. Unfortunately, this is a thorny problem that has no ideal solution. Indeed the authors of the landmark paper on the thinning of the Arctic perennial pack  (Rothrock Yu & Maykut, GRL 26, 1999) made a similar remark and took an approach similar to my own, using a 40-year climatology spatially averaged across much of the Canada Basin. However in contrast to my correction, they lacked snow-depth data to account for the strong influence of snow on the thermodynamic growth and ablation of sea ice. They state:

> This [seasonal] cycle was derived from an ice-ocean model with a 12-category ice thickness distribution [Zhang et al., 1998]. The model was forced with 40 years of winds and temperatures from the NCEP reanalysis, and thickness was averaged over those 40 years, over the SCICEX data release area …, and over the thickness distribution (including open water)

Their adjustment of measurements to a reference date of September 15 considerably changed mean thickness values, by as much as 0.6 m.

Snow depth exerts a dominant control on the thickening of sea ice during winter (see Maykut and Untersteiner, JGR, 1971; Dumas Carmack Melling, Cold Reg Sci Tech, 2005). Unfortunately, it varies appreciably with location and neither of the available long-term sites is close to Penny Strait; Resolute Bay is 230 km to the SSE and Eureka 520 km to the NE. It is my view that an ensemble of values measured over a long period can provide a useful measure of natural variability that is not available using one winter's data from a specific location. It goes without saying that a single winter's snow-depth development on sea ice at Resolute Bay would be of little relevance to that on ice in Penny Strait. I am presuming that the likelihood is greater for the climatology values.

I will undertake new seasonal "correction" calculations that incorporate the range of inter-annual variation in surface air temperature and snow depth, and will devise means to depict this as uncertainty on the figures that show seasonally "corrected data. The figures presently depict only the result of calculations based on long-term mean monthly values.

Line 289: I suggest to mark Resolute Bay and Eureka on Figure 1.

Yes, I can do this. Neither station falls within the border of the map, but both are close to it. These stations were chosen as the closest with the necessary long records of snow depth on sea ice.

Line 296-300: Is the uncertainty here significantly small compared to the difference of thickness from 1970s discussed later?

This comment joins that referenced to lines 286-292 in urging me to include an estimate of uncertainty in the seasonal "corrections". I can answer this question when I have done this.

Line 314-316: How is the numbers shown here sensitive to the assumption (half populated by multi-year ice)? Is the 'half populated by multi-year ice' the ice situation in the comparison period (1970s)?

Certainly there is a sensitivity since the MYI here is appreciably thicker than the 2.3 m attained by first-year ice. However, it is necessary to choose some selection criterion. One is therefore faced with the usual challenge when parsing data, which is a loss of degrees of freedom (and therefore increased uncertainty) as one imposes tighter constraints. It would certainly be possible to explore the domain with thresholds of 40% or 60% old ice. However, I suggest that the 50% threshold already allows a possible low bias from up to 50% seasonal ice that is already making my premise of "little change in multi-year ice thickness" more difficult to demonstrate.

Line 317: "the area north-west of Penny Strait" is ambiguous. I would suggest to show tracks of 1970s survey in a map for clarity.

The 1970s tracks are too numerous to show clearly on an existing map and I am very reluctant to add to an already long paper. I propose to state in the text the distance to the centroids of prior data used, and refer the reader to fig. 7 of my 2002 paper for details.

Line 318: The standard deviation of each mean thickness in the 1970s should be also shown in Figure 12.

I agree. This figure will be revised.

Line 373 – 394: Though the mechanism of thick ice formation described here is plausible, it is not shown that mechanical forcing on ice pack has not been changed since 1970s. In order to strengthen the argument, I suggest to show that statistics of wind forcing (e.g., strength, variance) has not been changed between the two comparison period or to show statistics of buoy tracks (e.g., onshore drift speed) has not been changed.

I take the reviewers point, but this is not a practical suggestion given the present state of knowledge. Wind forcing does not in itself form ridges; it is the mechanical failure of ice and the piling of fractured ice in response to the force of wind that does. Regrettably, quantitative understanding of the mechanics of ice ridging is not up to the inter-decadal task proposed by the reviewer. Even accurate estimates of wind stress in this area, present and past, are plagued by lack of observations, not only of the basic field of sea-level pressure, but also of the polar inversion that suppresses turbulence-mediated stress transmission down through the atmospheric boundary layer.

Line 431: Table 4 → Table 3. Thank you

Line 455: 'commo n' → common. Thank you

Line 461 – 462: 'ice' is repeated before and after the bracket. Thank you

Line 473: This sentence seems to me a bit strange. Probably colon or semicolon could be used to split the sentence. Yes. It is missing a "because". Thank you

Line 485 – 486, "The build-up of pressure sufficient to build large ridges occurs within a few hundred kilometres of coastlines during strong onshore wind storms". I do find neither an analysis nor time series supporting this sentence in this manuscript.

The reviewer is correct. Although we know from the earliest days of Arctic navigation that pressure ridges form when ice is under irresistible pressure, this paper has examined only the kinematic response of the pack ice, not the dynamical forcing. I will modify the statement to something along the lines of "The extreme deformation of pack ice required to build large ridges occurs within a few hundred kilometres of coastlines during strong onshore wind storms".

---

## Referee Report (RR1)

Review of "Sea-ice thickness on the northern Canadian polar-shelf: A second look after 40 years"

By: H. Melling

Submitted to The Cryosphere – tc-2021-393

This is my second review of this paper. I'd like to thank the author for addressing my previous comments and thoroughly revising and updating the manuscript. The paper provides a unique view of sea ice in a very remote area, but expands to look at the importance of dynamics in the formation of multiyear ice, which I found really interesting. My major concerns have all been addressed and I list only a handful of minor comments below.

Minor comments:

Line 8-10: Can this sentence be revised to clarify that data was collected from August 31 to December 10, and the instrument was recovered in May. It might help provide more detail than the general "winter 2009-2010".

Line 16: revise to read " …. When prior data are available *for comparison*".

Line 21-22: Thank you for adding in this part to clarify that MYI area has declined.

Line 47-50: consider revising to read something like "…because they are estimates from satellite altimeters that suffer from issues relating to snow depth, infrequent measures of sea level height, and a relatively large footprint (20-70 m diameter); in situ validation has been minimal (Kwok et al., 2020)."

Line 59: Consider revising to "More recent studies of ice in this area have focused on…"

Line 86: remove "become".

Line 103: replace "in-drift" with "import"

Line 108: "… when *the* Arctic climate…"

Line 131-132: Were the IPS ever deployed in Byam Martin Channel? Perhaps this text can be updated and suggested as an area of future work in the Discussion section.

Line 154: "…south-western side *of* Penny Strait…"

Line 185-186: Can you provide examples in brackets of what moderate and weak drift speeds are?

Line 213 and 215: revise to "southeasterly winds" and "northwesterly winds".

Line 238: Suggest adding "… and the assumption of no snow".

Line 364: revise to "… minimal change in multiyear ice *thickness* here during the last four decades". Population is a little misleading as there is less MYI than previously.

Line 372: It would be worth adding reference here to Kacimi and Kwok (2022) who recently showed amplified thinning of multiyear ice in the Arctic Ocean over the ICESat-2 time period (2018-2021). This would show greater thinning since the Kwok and Untersteiner work in 2011.
- Kacimi, S., & Kwok, R. (2022), Arctic snow depth, ice thickness, and volume from ICESat-2 and CryoSat-2:2018–2021. Geophysical Research Letters, 49, e2021GL097448. https://doi.org/10.1029/2021GL097448

Line 461: Remove "the" from "… area of ice within *the* which was thicker than 4 m…"

Line 504: replace "within" with "downstream of".

Line 524-526: Can these two types be referred to qualitatively as well as quantitatively based on thickness and chord length? Perhaps something like: Type 1, large conglomerate pans of MYI, and Type 2, smaller fractured ridges or rubble fields.

Line 530: suggest replacing "thermal wastage" with "loss".

Line 539-541: I really like this revised conclusions section. I think it expands the implications of this work, but I would suggest revising the last part of the centence to read "hundreds of kilometers in down-drift directions across the Canadian Polar Shelf and in the Beaufort Sea (via the Beaufort Gyre), Baffin Bay (via Nares Strait) and Greenland Sea (via Fram Strait).

Line 541: replace "they" with "the".
* * *
David Babb
University of Manitoba

---

## Author Response (AR2)

**Review of "Sea-ice thickness on the northern Canadian polar-shelf: A second look after 40 years"**

By: H. Melling

Submitted to The Cryosphere – tc-2021-393

Author's response to tc-2021-393-referee-report-2:

Line 8-10: Can this sentence be revised to clarify that data was collected from August 31 to December 10, and the instrument was recovered in May. It might help provide more detail than the general "winter 2009-2010".

> The text has been modified in response to this comment.

Line 16: revise to read " …. When prior data are available for comparison".

> The referee's suggested revision has been implemented.

Line 21-22: Thank you for adding in this part to clarify that MYI area has declined.

> No response necessary.

Line 47-50: consider revising to read something like "…because they are estimates from satellite altimeters that suffer from issues relating to snow depth, infrequent measures of sea level height, and a relatively large footprint (20-70 m diameter); in situ validation has been minimal (Kwok et al., 2020)."

> The text has been modified in response to this comment.

Line 59: Consider revising to "More recent studies of ice in this area have focused on…"

> The referee's suggested revision has been implemented.

Line 86: remove "become".

> Done.

Line 103: replace "in-drift" with "import"

> "in-drift" has been replaced by "inflow" .

Line 108: "… when the Arctic climate…"

> Done.

Line 131-132: Were the IPS ever deployed in Byam Martin Channel? Perhaps this text can be updated and suggested as an area of future work in the Discussion section.

> The sentences "We chose Penny Strait for the first year of 130 observation with an eye further west on Byam Martin Channel in future years. Because inter-annual variation was likely to be appreciable, we planned to sustain observations over several years" have been replaced with:
>
> "We chose Penny Strait for the first year of observation. Comparable data have since been collected elsewhere along the margin of the last ice area, in Byam Martin Channel and in Nares Strait. Because inter-annual variation was likely to be appreciable, we sustained observations in these channels for several years".

Line 154: "…south-western side of Penny Strait…"

> Done.

Line 185-186: Can you provide examples in brackets of what moderate and weak drift speeds are?

> This sentence actually documents the speed of wind, not ice. The referee's recommendation has been implemented. At the same time a previously overlooked error in the documentation of directions has been corrected.

Line 213 and 215: revise to "southeasterly winds" and "northwesterly winds".

> Done.

Line 238: Suggest adding "… and the assumption of no snow".

> The text has been modified in response to this comment.

Line 364: revise to "... minimal change in multiyear ice thickness here during the last four decades". Population is a little misleading as there is less MYI than previously.

Done.

Line 372: It would be worth adding reference here to Kacimi and Kwok (2022) who recently showed amplified thinning of multiyear ice in the Arctic Ocean over the ICESat-2 time period (2018-2021). This would show greater thinning since the Kwok and Untersteiner work in 2011.

Kacimi, S., & Kwok, R. (2022), Arctic snow depth, ice thickness, and volume from ICESat-2 and CryoSat-2:2018–2021. Geophysical Research Letters, 49, e2021GL097448. https://doi.org/10.1029/2021GL097448

I have chosen not to adopt this suggestion by the referee. My rationale is that the Kwok and Untersteiner paper is that most relevant to the 2009 observations presented in my paper. The 2022 paper would of course have been the best if the sonar data were available for the years 2018-2021 which are the focus of the paper by Kacimi and Kwok.

Line 461: Remove "the" from "... area of ice within the which was thicker than 4 m..."

The referee's suggestion has drawn my attention to this unclear sentence, although the correction suggested was not approriate. The text has been revised.

Line 504: replace "within" with "downstream of".

This suggestion depends on how the geographic extent of the "last ice area" is defined, which is vaguely. In reality the enigmatic nomenclature ensures that, to borrow from Joni Mitchell, "we won't know what we have got 'til it's gone". Clearly, any definition is time dependent. I chose to consider that, at present, the "last ice area" is the domain in which multi-year ice is dominant. Therefore, the observing site is, at present, at the southern margin of the last ice area. I have adjusted the text accordingly.

Line 524-526: Can these two types be referred to qualitatively as well as quantitatively based on thickness and chord length? Perhaps something like: Type 1, large conglomerate pans of MYI, and Type 2, smaller fractured ridges or rubble fields.

The text has been modified in response to this comment.

Line 530: suggest replacing "thermal wastage" with "loss".

I have replaced "wastage" with "ablation".

Line 539-541: I really like this revised conclusions section. I think it expands the implications of this work, but I would suggest revising the last part of the sentence to read "hundreds of kilometers in down-drift directions across the Canadian Polar Shelf and in the Beaufort Sea (via the Beaufort Gyre), Baffin Bay (via Nares Strait) and Greenland Sea (via Fram Strait).

The text has been modified in response to this comment.

Line 541: replace "they" with "the".

Done.